# Automated Process Planning for Embossing and Functionally Grading Materials via Site-Specific Control in Large-Format Metal-Based Additive Manufacturing

**DOI:** 10.3390/ma15124152

**Published:** 2022-06-11

**Authors:** Michael Borish, Brian T. Gibson, Cameron Adkins, Paritosh Mhatre

**Affiliations:** Oak Ridge National Laboratory, Knoxville, TN 37932, USA; gibsonbt@ornl.gov (B.T.G.); cadkin17@vols.utk.edu (C.A.); mhatreps@ornl.gov (P.M.)

**Keywords:** slicing, embossing, additive manufacturing, large-format DED

## Abstract

The potential for site-specific, process-parameter control is an attribute of additive manufacturing (AM) that makes it highly attractive as a manufacturing process. The research interest in the functionally grading material properties of numerous AM processes has been high for years. However, one of the issues that slows developmental progress in this area is process planning. It is not uncommon for manual programming methods and bespoke solutions to be utilized for site-specific control efforts. This article presents the development of slicing software that contains a fully automated process planning approach for enabling through-thickness, process-parameter control for a range of AM processes. The technique includes the use of parent and child geometries for controlling the locations of site-specific parameters, which are overlayed onto unmodified toolpaths, i.e., a vector-based planning approach is used in which additional information, such as melt pool size for large-scale metal AM processes, is assigned to the vectors. This technique has the potential for macro- and micro-structural modifications to printed objects. A proof-of-principle experiment is highlighted in which this technique was used to generate dynamic bead geometries that were deposited to induce a novel surface embossing effect, and additional software examples are presented that highlight software support for more complex objects.

## 1. Introduction

Slicing is a necessary and critical step in the additive manufacturing (AM) workflow. Because it is commonplace for off-the-shelf or Original Equipment Manufacturer (OEM)—provided slicers to be utilized, slicing is an under-researched aspect of AM. However, a slicer that is well tailored to a specific AM machine and process can make the differences between a process that is simply adequate and one that achieves its full potential for productivity, quality, and site-specific process-parameter control; it can also mean that full automation of process-planning activities that otherwise would be very tedious and time-consuming can be realized.

Before slicing can begin, an object is typically converted from CAD (computer aided design) to an STL (stereolithography) file. STL represents all surface geometry using triangles [1]. This STL is then intersected by a horizontal plane at incremental heights corresponding to the preferred layer height of construction on the AM machine. These intersections create a stack of 2D polygons corresponding to the layers that must be filled. This filling takes the form of several types of paths, including perimeters, insets, skins, and infill. These styles of pathing may include both closed contours and open polylines.

However, the generation of an object does not need to be limited to the contours of 2D polygons. Indeed, various authors have investigated the generation of features on or within an object for goals such as security or quality assurance [2,3]. To affect such a goal for large-format metal-based AM systems, site-specific control is necessary. Site-specific control has previously been investigated for numerous reasons, including process stability and microstructure control.

Various works have investigated the use of features on or within objects for the purpose of security or quality assurance [2,3]. In this referenced work, features are built into the object as part of the CAD design process and are later evaluated via visual inspection. The embossing process presented here can accomplish similar behavior, but as multiple objects are accounted for as part of the pathing algorithm, there is no need to spend time designing the features in CAD.

A significant body of work has also investigated moving beyond the confines of layers and geometric boundaries. So-called out-of-bounds printing was the focus of work by Patricio et al. [4]. This work sought the freeform construction of objects by utilizing a supporting medium for the construction material.

Some work has also sought to modify what constitutes a layer. In an investigation by Alkadi et al., conformal mapping was used to construct objects on a curved surface [5]. Using conformal mapping, the object was warped to match the surface it would be constructed on. Interest is continuing to grow in conformal slicing for supporting greater geometric complexity and improved part quality [6,7,8,9]. There has also been significant interest in voxel-based process planning, a method that lends itself well to localized or site-specific parameter modifications [10,11,12,13,14,15], but does not necessarily lend itself well to the way in which many AM machines traditionally operate; vector-based processes planning is more widely applicable, and in the present work, a vector-based, site-specific process planning technique is presented.

In addition to extending the concepts of layers and moving beyond the typical geometric analysis, some works have also investigated site-specific control. Dehoff et al., Lee et al., Shi et al., and Raghavan et al. are four such examples [16,17,18,19]. This body of work focuses on metal powder systems where site specific control via open-loop modifications to laser power, focus, speed, etc., are used to affect the object’s microstructure. Indeed, in the mentioned work, the authors were concerned with grain growth and orientation in the resulting microstructure, one of the primary outcomes of interest in AM that would stem from the use of a site-specific control [20,21,22].

A noteworthy use of a site-specific control in stereolithography is the two-color photopolymerization of tailored resins for inducing varying region thicknesses and topographical patterning [23].

Site-specific control of the melt pool for macrostructure control in a large-scale direct energy deposition (DED) system has also been investigated. Heralic et al. [24] and Nassar et al. [25] have been significant contributors in the area of DED process control. Melt pool manipulation and control has been investigated through numerous methodologies, including modeling and mapping approaches [26,27,28,29] and closed-loop feedback control for achieving uniform deposition properties [30,31,32,33], which was the prior state-of-the-art. Multiple works by Gibson et al. have demonstrated melt pool monitoring and closed-loop control [34,35,36]. In this body of work, the authors demonstrated melt pool control via size and shape manipulation, and its resulting impact on the construction process.

A novel extension of this work was a manually programmed embossing process carried out via site-specific, closed-loop control [37], in which dynamic bead geometries were generated and deposited along an unmodified toolpath, resulting in a 2D image embossed on the side of a 3D printed part. This method is in contrast with other efforts that require the toolpath to be modified to induce, or work in coordination with, dynamic bead geometries [38,39,40]. This was a first-of-its-kind demonstration of through-thickness, closed-loop, site-specific control in DED [41], and further exploration of this method demands that the process planning be automated. Accordingly, the focus of the present work is the automation of the process planning required to implement this method, and it is believed that there are numerous benefits and advantages associated with it, which include the following:Through-thickness, site-specific, process parameter control for many AM processes, including metal powder bed fusion (not just DED);Fully automated process planning, rather than time-consuming manual programming or the development of bespoke solutions;Macro-structural control for achieving dynamic bead geometries, the benefits of which include volumetric defect mitigation and embossing for object security;Micro-structural control for achieving functionally-graded part properties.

The automation of the process planning method will be presented with software examples showing techniques for infill modification and embossing on both flat and curved surfaces, which is the realization of a concept for generating features outside of the pathing definitions, a novel approach for slicing in AM, which makes use of primary (parent) and secondary (child) geometries.

## 2. Materials and Methods

To facilitate the new embossing process, site-specific control is necessary. As a result, the ORNL Slicer 2.0 (Oak Ridge National Laboratory, Knoxville, TN, USA) was augmented to provide site-specific capabilities. In Slicer 2, the concept of a mesh was extended to allow for additional control, and a mesh can be designated as one of several different types. The types of interest for this discussion are build and settings meshes. Build meshes are the typical mesh that is loaded, cross-sectioned, and has pathing laid out. Settings meshes, in comparison, do not generate any pathing, but rather modify specific regions of a build mesh. A simple example for a polymer-based approach is shown in Figure 1.

In this example, on the left, the infill density is modified to be denser near the holes to provide additional support when the holes are later bored. On the right of the figure is the opposite example, where an area is prevented from having any infill pathing. In addition to infill density, settings meshes allow for the modification of numerous other physical parameters such as the extrusion rate and power. Manipulation of the input power was crucial for the large-format DED system that the embossing process was applied to.

With the ability to modify settings for an arbitrary area in place, the embossing process was developed as a logical extension of that capability. The first step is to allow the ingestion of multiple STLs in combination to form a parent–child relationship. This relationship defines which STL contributes to build pathing or settings modifications that result in out-of-contour pathing. The parenting system also allows for easy manipulation of the objects. An example visualization is shown in Figure 2.

As part of this loading process, the child mesh can be manipulated similarly to any other mesh. This manipulation includes typical affine transformations such as translation, rotation, and scaling to fit the embossed mesh appropriately to the surface of its parent.

To be clear, the parenting system in Slicer 2 is intended for user convenience. When a child mesh is parented, any manipulation of the parent with respect to translation, rotation, and scaling is automatically applied to the child as well. Additionally, the largest surface of each mesh is automatically aligned. However, final adjustments must be made by the user. The most important adjustment is how deeply the child should be embedded in the parent. How deeply a child object can be embedded is a function of its design. The child object should be designed to have a thickness of at least one bead width. As will be shown in subsequent pictures, the cross-section of the child intersecting the parent’s pathing is what gives rise to the emboss behavior. Thus, the thicker the child mesh, the deeper it can be embedded in its parent. By embedding the child deeper, the cross-section of the child will intersect more pathing of the parent mesh. Increased path intersections produce more sites for melt pool expansion and can potentially make the embossed feature more pronounced.

With the two meshes loaded for construction, slicing can begin. Each step of slicing involves one of several foundational algorithms, such as plane-object intersection, polygon offsetting, skeleton generation, and line–polygon intersection. A full explanation of each algorithm of the slicing process is outside the scope of this paper. However, the computational library Computational Geometry Algorithms Library (CGAL) is one of multiple detailed sources of these algorithms [42].

The parent mesh is first cross sectioned to determine the appropriate area for pathing. In addition, the child mesh meant for embossing is also cross-sectioned. This secondary cross-section is bundled with additional pieces of spatial information, including the cross-sectioning plane’s orientation. This orientation is important to facilitate the process for an arbitrary slicing plane. For ease of explanation, the example will assume a standard horizontal slicing plane that moves up the *Z* axis with the entire child mesh being embossed on a single flat surface. However, there is nothing that prevents this approach from working with an arbitrary slicing plane or curved surfaces. An example of this cross-sectioning is shown in Figure 3.

Figure 4 shows an example of the cross-sectioning. For this example, the yellow slice plane begins moving up the *Z* axis. For the first layer, the cross-sections in the bottom picture are the result. The larger rectangles represent the wall (the parent object) while the smaller rectangle represents the stem of the left (the child object). In this example, the wall is two beads thick, which results in the two large rectangles that will be filled with appropriate pathing in the subsequent steps. The child cross-section is carried along during this process and the process repeats for all layers.

Once the cross-sectioning has been completed, the pathing for the parent object proceeds as normal. That is, all appropriate contours in the form of perimeters, insets, skin, infill, and skeletons are applied. An example of the resultant pathing is shown below.

The parent object then moves into a post-processing stage where modifications related to the child object are applied. Recall that the cross-section for the child object is passed along during this process and that this cross-section is the result of the child object protruding from the parent object. As a result, simple geometric logic in the form of line–polygon intersection can be applied. Line–polygon intersection can be decomposed into line segment–line segment intersection and can be computed via the Bentley–Ottmann algorithm [43]. This algorithm is a line sweep algorithm that sweeps a line across all segments that have been sorted by their x location. This algorithm results in a set of intersections. A visualization of the pathing with the child object’s cross-section is shown in Figure 5.

In this example, the focus will be on the first path of the parent object as it is the one that intersects the child object. If the pathing from the parent object intersects the cross-section of the child object, the segment in that path is broken into multiple smaller segments. In this example, the path would be broken into three subsegments with each segment assigned appropriate physical parameters upon conversion to G-code commands. An example of the G-code commands for the three segments above is shown below in Figure 6:

The physical parameter of interest is a melt pool size set-point, defined as ESP. In these commands, the typical variables that indicate linear motion are unchanged and ESP is simply concatenated to the end. ESP is in reference to the melt pool control system attached to the machine. This control system is a camera-based inspection system that enables control of the melt pool via modulation of the laser power. With the necessary control support, this variable is routed to the system for the melt pool control.

In this example, ESP represents a percentage change from nominal. So, a G-code command including ESP1.0 is the typical build command with the melt pool at a nominal size. The second line would have a melt pool 50% larger than nominal.

Thus, as the object is constructed via the pathing of the parent object, subsegments of the pathing will have modulation information. This modulation has the practical effect of expanding or contracting the melt pool. The expansion or contraction gives rise to the child object embedded in the face of the parent object without explicit pathing.

As mentioned previously, in the work by Gibson et al., melt pool control via laser power was described for a large-format DED system. As a result of this work, a power modulation setting is included as part of the slicing software. In conjunction with other standard user settings indicating appropriate build velocities and bead width requirements, the power modulation allows for a simple percentage change calculation. With this algorithm in place, a demonstration piece was constructed to showcase the embossing technique.

## 3. Results

Utilizing the algorithm described in the last section, an example embossed object was constructed. This object was constructed on a wire fed large-format DED system. The feedstock used was a Ti-6Al-4V wire with a 1.6 mm diameter. The parent geometry selected was a double-bead wall (175 mm in length and 150 mm in height). At the given layer height, the wall was 94 layers. This parent geometry was constructed via skeleton paths that represent open-loop polylines. These polylines are represented as two open paths that are parallel to one another. The child geometry that was selected was ORNL’s oak leaf as it represents an object with non-trivial complexity. A visualization of the two objects and the resultant pathing is shown in Figure 3 and Figure 4, respectively. The physical demonstration piece for this visualization is shown in Figure 7.

As mentioned in the previous section, there is nothing inherent in the process that precludes the application of this approach to objects with different build planes or non-flat surfaces. The algorithm described in the previous section was applied to a few additional examples for verification. These examples are shown in Figure 8.

## 4. Discussion

As can be seen from Figure 7, the automated implementation for embossing was successful. After post-processing of the build is complete, the oak leaf is clearly visible on the front face of the wall. To affect this result without this algorithm, this DED process, which would normally be constrained by a specific set of design rules that typically require a lower resolution component detail [44]. Full details of the physical parameters can be found in the previous work conducted by Gibson et al. [37].

Additionally, as shown in Figure 8, a child object can be embossed upon its parent, regardless of the curvature of its surface. The example of the left is a hexagon embossed onto the bottom of a single piece chair, which represents a large-scale application. In contrast, the example on the right is the oak leaf embossed on the side of the tugboat STL, which is commonly used in small-scale printing.

The algorithmic approach presented in Section 2 requires additional computation beyond the typical slicing of the parent object. For this discussion, the computational cost of the algorithms will be described with Big O notation. Big O notation is used to classify algorithms according to how their run time or space requirements grow as the input size grows. There are two major algorithms for the presented approach: the cross-sectioning of the child object and the testing of the parent pathing for intersections with the child cross-section. The cross-sectioning of the child object requires evaluation of the slicing planes with each of the facets of the child mesh, just as would happen for the parent mesh. In terms of computational cost, this results in O(kn) complexity, where k is the number of layers and n is the number of triangles representing the mesh in the typical STL format.

The testing of the parent pathing is accomplished via the Bentley–Ottmann algorithm. The Bentley–Ottmann algorithm processes a sequence of 2n + k events, where n denotes the number of input line segments and k denotes the number of crossings. Each event requires log n time to evaluate. Ignoring constants, the complexity is thus O ((n + k) log n).

Practically, the embossing algorithm requires little additional computation. Nearly all of the necessary computational infrastructure and memory allocation is already handled as part of the slicing process of the parent object. Slicing times for objects can vary significantly, but the typical slicing times for the type of parent objects presented here are less than a minute. The slicing for the wall used for the demonstration can be completed in a matter of seconds on a standard desktop workstation. The additional impact of the computation of the child object is relatively small and only increases the slicing time by a few seconds in most cases.

The computational impact of the child object processing could be more significant depending on its design. Recall in Section 2 that the child should be designed to have a thickness of at least one bead width. This is to guarantee the child object intersects with the pathing of the parent object. However, a child object could be designed to be very thick, potentially thicker than the parent, which would result in significantly more pathing, creating intersections between the parent and child object. In this case, the computational cost would grow and would require balance with the design. Generally, with a greater number of intersections and thus melt pool modulations, the more pronounced the embossing effect.

With respect to the use of features for security or quality assurance, the typical approach involves the explicit design of the said feature in CAD. This design can be time consuming and is unique to the object under consideration. Using the algorithm described here, the feature could be designed once, exported, and reused for any number of parent objects. This could reduce the amount of time spent designing objects. Ultimately, the generation and embedding of certain features, such as a barcode, could be handled automatically by the algorithm.

## 5. Conclusions

A new approach for out-of-bounds printing via site-specific control, and the full software automation thereof, was presented herein. This required the introduction of some novel concepts:The development of settings meshesThe manipulation of pathing via setting modification, rather than the creation of additional pathing

This capability can be paired with a parenting feature and appropriate hardware to allow for a second object to appear on the face of another without explicit pathing. A proof-of-principle demonstration object that was previously constructed was highlighted to validate this approach, and additional software examples were presented that highlighted the applicability of this process to objects with additional complexity. This capability has wide ranging implications in AM, including through-thickness, site-specific, process parameter control, macro-structural control for achieving dynamic bead geometries (benefits include volumetric defect mitigation and embossing for object security), and micro-structural control for achieving functionally-graded part properties.

The findings of the present work also point to additional future work that would be beneficial for the process planning approach. These areas include the following:Extending the algorithm to account for embossing discrepancies;An investigation into the concept of imbedded security measures into objects;Improved algorithm robustness to account for pronounced embossing and overhang features that may currently pose a challenge.

Furthermore, the methods presented herein are applicable to a wide range of AM processes, including metal powder bed fusion processes, and opportunities will be sought to apply the new fully automated process planning approach to site-specific control demonstrations in these processes.

## Figures and Tables

**Figure 1 materials-15-04152-f001:**
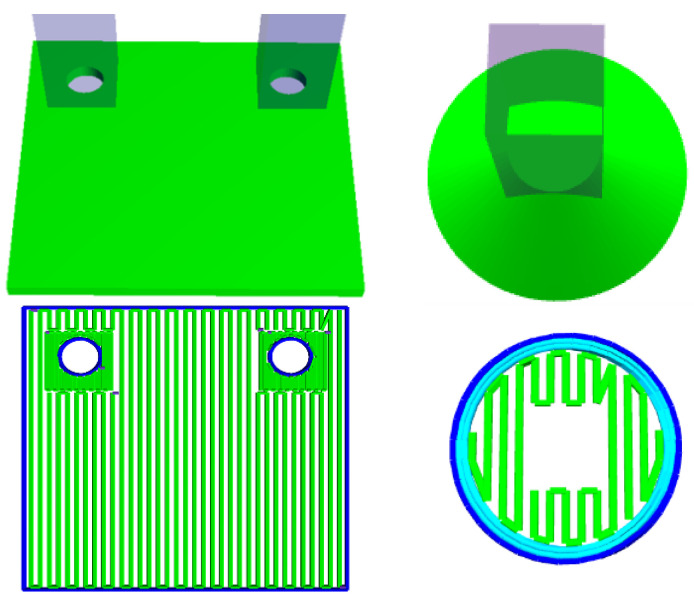
Example of site-specific control application. Left: two rectangles are placed around the holes of the object to densify the infill pattern. Right: Specific areas are overridden to have no infill.

**Figure 2 materials-15-04152-f002:**
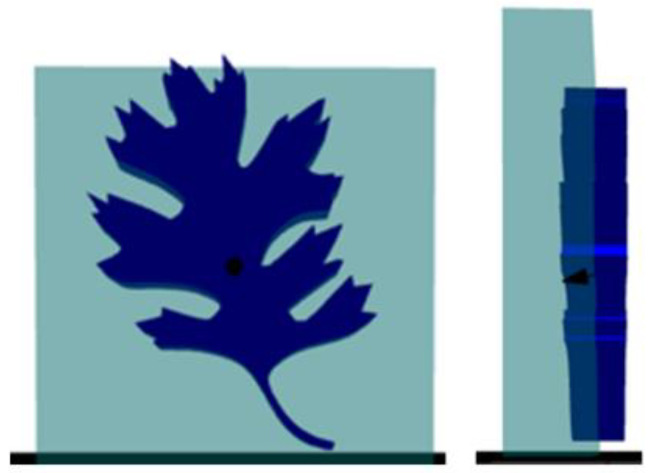
Example of two objects set as the parent and child. The wall (light blue) represents the parent, while the oak leaf (dark blue) represents the child.

**Figure 3 materials-15-04152-f003:**
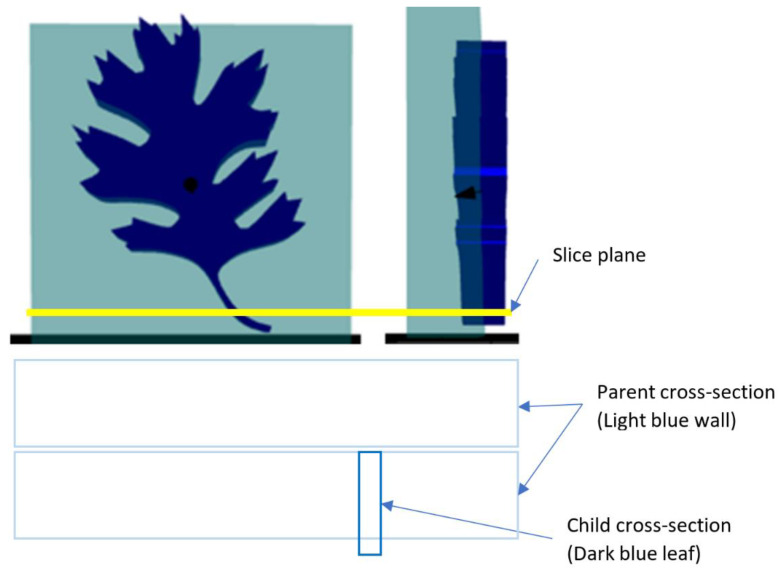
Example cross-section. **Top**: The yellow line represents the slicing plane. **Bottom**: The cross-section results in two rectangular areas for the wall and one smaller area for the leaf stem.

**Figure 4 materials-15-04152-f004:**
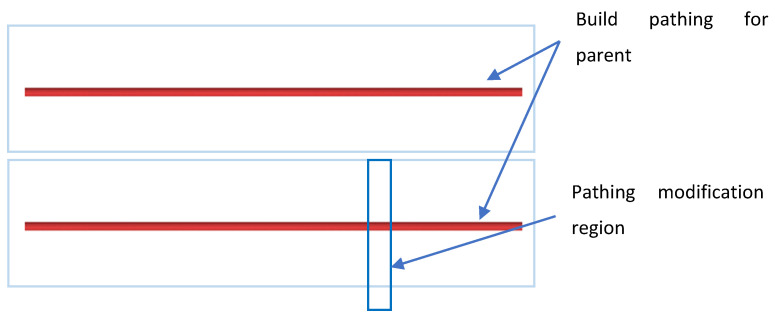
Resultant pathing. The red paths represent the build paths of the wall (parent).

**Figure 5 materials-15-04152-f005:**
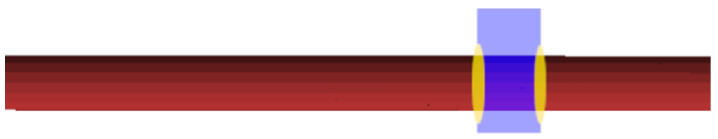
Zoomed in visualization of the first build path. Red represents the pathing for wall construction, blue is the cross-section of the oak leaf at its stem for its first layer, and yellow areas represent points of intersection.

**Figure 6 materials-15-04152-f006:**
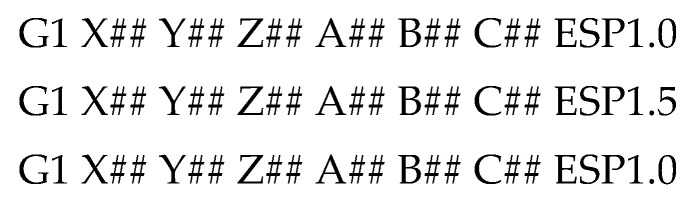
Example of G-code commands.

**Figure 7 materials-15-04152-f007:**
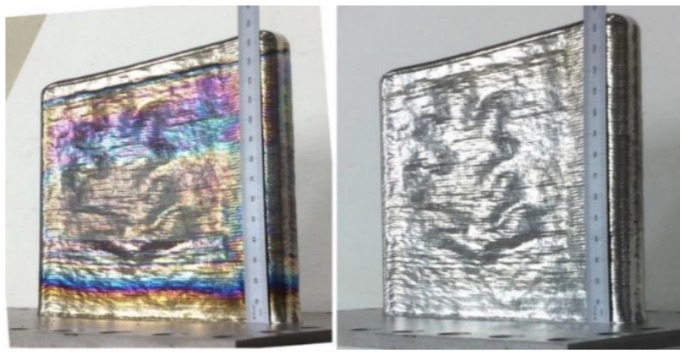
Demonstration object. **Left**: as constructed. **Right**: after post-processing heat treatment. Gibson et al. [37].

**Figure 8 materials-15-04152-f008:**
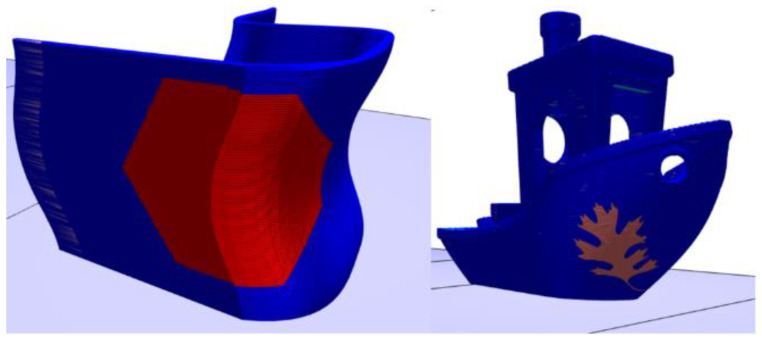
Examples of embossing on non-flat surfaces. **Left**: Hexagon on chair bottom. **Right**: Oak leaf on tugboat.

## Data Availability

Not applicable.

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
