# Peer review of "Automated Process Planning for Embossing and Functionally Grading Materials via Site-Specific Control in Large-Format Metal-Based Additive Manufacturing"

_materials, 2022, doi:10.3390/ma15124152_

Round 1

Reviewer 1 Report

Some results were presented, while the scientific contribution of this work was limited. I am very sorry but I can’t recommend this paper.

  1. There are no micrographs, macrographs, equations or schematic diagrams presented to enable one to further evaluate the depth of the work. Just a couple of figures shown the examples, of course are not sufficient for a paper to be entitled for publication in an international journal. The work is shallow and superficial.
  2. A general view of industrial applications of the embossing process together with the algorithm is necessary to be presented. For instance, if somebody is not enough familiar to proposed embossing process, does not realize the advantage etc.
  3. The ”Materials and Methods” parts are quite vague and short with no details on how to conduct the embossing, how to modify what constitutes a layer and how to optimize the algorithm, etc. I can’t understand the details well.
  4. The abstract is not clear, the objectives and the main findings are not properly presented.
  5. The evidences provided for the claims propounded in Discussion are marginal and trivial.
  6. Despite English is generally good, in many instances it needs to be edited, reviewed and improved.

Reviewer 2 Report

  • Please refine (concise) the last paragraph of the Introduction section. It reflects to be Part of Methodology.
  • Please mention the novelty of the work at the end of Introduction section.
  • In construction, 2 objects must be set, one is PARENT, other is CHILD. How to design the child? It is not clear. As I read the manuscript, there may be a lot of options. Please elaborate.
  • Is this path for AM is in conceptual state?
  • Do you have any experimental support for this approach? Please explain.
  • In Discussion section, it is mentioned as “In comparison to other approaches, it would not be necessary to design these features for a specific object via CAD. Instead, these features could simply be loaded or generated automatically by the slicer to be embedded in the parent object”. What is the time required for processing in the new approach? Can you comment on this aspect?
  • The details and approach description given to understand the concept is too shallow. Technically, supporting explanation and documents are not sufficient.
  • In some of the REFERENCEs, the title of the paper, the first letter of each word is upper case whereas in other references only first letter of the first word is upper case letter. Please correct it for the Format uniformity.

Reviewer 3 Report

The authors have devised an algorithmic approach to automatically generate the necessary geometric slicing pathways in large format metal additive manufacturing of embossing and functionally graded materials. This work is interesting for powder metallurgy community. There are many loose ends in the discussion. Some comments can be useful to further improve this work.

  1. Introduction is poorly made. The authors should add recent trends in this area and compare the advantage of their work.
  2. The authors should cite some more relevant references about slicing technique and put emphasis on the advantage of this study.
  3. The presentation of figure 1 is ordinary. It should be improved.
  4. The authors refer Fig. 4 from reference [30]. Is this from the present work?
  5. The discussion section is weak in this manuscript.
  6. The authors frequently refer reference [30]. What is the highlight of this work then?
  7. The conclusion section looks like a summary or another abstract of this work.
  8. The important conclusion in this work can be bulleted.

Overall, authors need more work and analysis and better presentation of their work.

Round 2

Reviewer 1 Report

Frankly speaking I can't recommend publication. However, the authors have tried their best. Thus, I have to be charitable and give them one chance.

Reviewer 3 Report

The authors have answered all the comments.